# Experimental Research on Seepage Law and Migration Characteristics of Core-Shell Polymeric Nanoparticles Dispersion System in Porous Media

**DOI:** 10.3390/polym14091803

**Published:** 2022-04-28

**Authors:** Xiaohe Huang, Yuyi Wang, Yunqian Long, Jing Liu, Han Zheng, Wen Nie, Hongyan Han

**Affiliations:** 1School of Petrochemical Engineering & Environment, Zhejiang Ocean University, Zhoushan 316022, China; huangxh@zjou.edu.cn (X.H.); wangyuyi1210@163.com (Y.W.); zhenghan0413@163.com (H.Z.); niewen357@163.com (W.N.); 2United National-Local Engineering Laboratory of Harbor Oil & Gas Storage and Transportation Technology, Zhejiang Ocean University, Zhoushan 316022, China; 3Donghai Science and Technology College, Zhejiang Ocean University, Zhoushan 316022, China; 4Department of Construction Engineering, Hebei Vocational University of Industry and Technology, Shijiazhuang 050091, China

**Keywords:** seepage law, core–shell polymeric nanoparticles, microtubule flow, profile control, enhanced oil recovery

## Abstract

The nanoparticles dispersion system has complex migration characteristics and percolation law in porous media due to the interaction between the nanoparticles and porous media. In this paper, lab experiments were carried out to characterize the morphology, particle size distributions, and apparent viscosities of SiO_2_/P(MBAAm-*co*-AM) polymeric nanoparticle solution, investigate its migration characteristics in porous media, and probe its capability of enhanced oil recovery (EOR) in the reservoirs. Quartz microtubule, sand pack, and etched glass micromodels were used as the porous media in the flow and flooding experiments. Gray image-processing technology was applied to achieve oil saturation at different flooding stages in the micromodel for calculating the EOR of the SiO_2_/P(MBAAm-*co*-AM) polymeric nanoparticle solution. The results show that The SiO_2_/P(MBAAm-*co*-AM) polymeric nanoparticles are spherical with diameters ranging from 260 to 300 nm, and the thicknesses of the polymeric layers are in the range of 30–50 nm. As the swelling time increases from 24 to 120 h, the medium sizes of the SiO_2_/P(MBAAm-*co*-AM) polymeric nanoparticles increase from 584.45 to 1142.61 nm. The flow of the SiO_2_/P(MBAAm-*co*-AM) polymeric nanoparticles has obvious nonlinear characteristics and a prominent scale effect at a low-pressure gradient, and there should be an optimal matching relationship between its injection mass concentration and the channel size. The flow tests in the sand packs demonstrate that the SiO_2_/P(MBAAm-*co*-AM) polymeric nanoparticles can form effective plugging in the main flow channels at different permeability areas and can break through at the throat to fulfill the step-by-step profile control. Moreover, the profile control of the SiO_2_/P(MBAAm-*co*-AM) polymeric nanoparticles strengthens with an increase in their swelling time. The microscopic flooding experiment in the etched glass micromodel confirms that the SiO_2_/P(MBAAm-*co*-AM) polymeric nanoparticles can block dynamically and alternatively the channels of different sizes with the form of loose or dense networks to adjust the fluid flow diversion, improve the sweep efficiency, and recover more residual oil. The SiO_2_/P(MBAAm-*co*-AM) polymeric nanoparticles can achieve an enhanced oil recovery of 20.71% in the micromodel.

## 1. Introduction

At present, many oilfields have entered a high water-cut period after experiencing long-term waterflooding from their early development stage. The long-term erosion of the reservoir rocks caused by water injection enhances the heterogeneity of the reservoir. The dominant channels of the water flow are formed in the high permeability layers, and the injected water can transport rapidly into the production wells along with them [1,2]. A highway of the water flow is gradually formed, which reduces the producing degree of low and medium permeability layers [3,4]. In addition, due to the high viscosity of crude oil in the formation, the dominant channels will lead to the rapid rise of a water cut and decline of oil production after water breakthrough, which seriously affects the development effect of the oilfield [5,6]. Conventional polymer flooding can block the dominant channels in the reservoir to improve the oil recovery [7,8,9]. However, it is difficult for the polymer flooding to achieve good profile control due to the profile inversion and inaccessible pore volume. It is estimated that after water flooding, there is about 65–77% and 23–35% of the residual oil in the un-swept and swept areas, respectively [10]. Therefore, it is crucial for enhanced oil recovery (EOR) in the reservoir to recover the residual oil from the un-swept areas.

In recent years, more and more attention has been paid to the profile control of the polymeric nanoparticles (NPs) in the heterogeneous reservoir because the polymeric nanoparticles have a small initial particle size and good dispersion and expansion properties in aqueous solution [11,12,13,14]. The polymeric nanoparticles can absorb water to swell and form a plug in the reservoir after being injected into the formation. Thus, the subsequent liquid can be forced to change flow direction and enter the medium and low permeability layers, which improves the utilization degree of the medium and low permeability layers and recovers more residual oil from the un-swept areas [15,16,17]. At the same time, the polymeric nanoparticles can break through the pore throat by elastic deformation under the action of pressure and then migrate into the deep part of the reservoir, which can achieve the step-by-step profile control in the reservoir [18,19,20]. The polymeric nanoparticles with a shell of polymer and a core of inorganic nanoparticles, referred to as polymer-coated nanoparticles, have a more promising application in EOR compared to the pure polymeric nanoparticles because they can provide significant advantages such as increased salt and heat resistance, greater stability at oil and water interfaces, and stronger profile control in porous media [21,22].

Some studies have confirmed that polymer-coated nanoparticles can improve oil recovery in heterogeneous reservoirs. Khalil et al. investigated the EOR ability of surface-functionalization superparamagnetic nanoparticles (SPNs) using a sand pack [23]. The nanofluid including the SPNs-PAM (polyacrylamide) nanoparticles achieved the highest secondary oil recovery of 19.28% original oil in place (OOIP). Bila et al. investigated the EOR potential of polymer-coated silica nanoparticles (PSiNPs), and the results showed that the secondary and tertiary recoveries by the PSiNPs ranged from 60 to 72% OOIP and from 2.6 to 5.2% OOIP, respectively [24]. Lai et al. synthesized nano-SiO_2_ hyperbranched copolymer (HPBS) using an in situ free radical polymerization reaction and the HPBS improved the oil recovery by 18.74% after a waterflood [25]. Behzadi et al. evaluated the secondary EOR ability of polymer-coated silica NPs using an etched glass micromodel and found that the polymer-coated silica NPs achieved 59% OOIP in comparison with 39% OOIP by waterflood [26]. In previous studies, we used the distillation precipitation polymerization method to successfully prepare SiO_2_/P(MBAAm-*co*-AM) polymeric nanoparticles, and the flooding tests demonstrated that the SiO_2_/P(MBAAm-*co*-AM) polymeric nanoparticles could improve oil recovery from 10.28 to 21.97% by increasing the swelled particle sizes from 580 to 1160 nm [27]. The existing studies mainly focus on the secondary and tertiary EOR ability of different polymer-coated nanoparticles to obtain an oil displacement agent with great potential. There is very little research on the EOR mechanisms of the polymer-coated nanoparticles. Choi et al. attributed the EOR mechanisms of the polymer-coated silica NPs to changing the wettability of the rock surface and reducing the interfacial tension (IFT) between oil and water [28]. Yue et al. thought that the polymer-coated nanoparticles could block the large pore channels of different sizes to achieve step-by-step profile control, leading to improved swept volume to enhance oil recovery [29]. It can be seen that there is no unified understanding of the EOR mechanisms of the polymer-coated nanoparticles. The seepage law and migration characteristics of the polymer-coated nanoparticles are closely related to their EOR mechanism. Thus, it is necessary for better understanding of the EOR mechanisms of the polymer-coated nanoparticles to investigate their seepage law and migration characteristics in porous media.

In this study, we prepared the SiO_2_/P(MBAAm-*co*-AM) polymeric nanoparticles based on our previous research [27]. The morphology, particle size distributions, and apparent viscosities of the SiO_2_/P(MBAAm-*co*-AM) polymeric nanoparticle solution were determined using transmission electron microscopy (TEM), dynamic light scattering (DLS), and a viscometer, respectively. To investigate the seepage law, migration characteristics, and EOR ability of the SiO_2_/P(MBAAm-*co*-AM) polymeric nanoparticle solution, quartz microtubules, sand packs, and etched glass micromodels were used as the porous media to carry out the microtubule flow experiments, profile control experiments, and microscopic flooding experiments. The image-processing technology and a software analysis method were used to deal with the microscopic flooding images for quantitatively calculating the EOR of the SiO_2_/P(MBAAm-*co*-AM) polymeric nanoparticle solution. This study will provide a new method to investigate the seepage law of the nanofluid by carrying out microtubule flow experiments and will be beneficial for comprehensively understanding the flow characteristics of polymeric nanoparticles in porous media. Simultaneously, we expect that the study can provide a reference for formulating an injection strategy for the application of polymeric nanoparticles in mature oilfields.

## 2. Materials and Methods

### 2.1. Materials

Tetraethyl orthosilicate (TEOS, 28.5%), acrylamide (AM), *N*,*N*-methylene-bis-acrylamide (MBAAm), azobisisobutyronitrile (AIBN), acetonitrile, ethyl alcohol, and ammonia solution (25%) were purchased from Sinopharm Chemical Reagent Co., Ltd., Beijing, China, for the synthesis of SiO_2_/P(MBAAm-*co*-AM) polymeric nanoparticles. Calcium chloride, magnesium chloride, sodium chloride, potassium chloride, sodium sulfate, sodium carbonate, and sodium bicarbonate were purchased from Sinopharm Chemical Reagent Co., Ltd., Beijing, China, for the preparation of simulated formation water. All the chemical reagents were of analytical grade, and all the solutions used in this work were freshly prepared using deionized water. The experimental gas purchased from a local supplier (Zhejiang Hongtai Biotechnology Co., Ltd., Zhoushan, China) was nitrogen with a purity greater than 99.999%. The synthetic formation water was a brine with a total dissolved solids (TDS) value of 5260 ppm for all flow tests. The crude oil sample obtained from a Daqing oil reservoir had a density of 0.885 g/cm^3^ and a viscosity of 10.2 mPa s (60 °C) and was used to carry out the microscopic flooding experiment.

### 2.2. Synthesis of SiO_2_/P(MBAAm-co-AM) Polymeric Nanoparticles

TEOS in an amount of 11.2 mL, ethyl alcohol in an amount of 84.1 mL, and deionized water in an amount of 17.1 mL were mixed in a 250 mL conical flask. Then, 4.76 mL of ammonia was added to the mixed solution under the condition of continuous stirring at 500 rpm. The mixed solution was reacted for 6 h under the condition of continuous stirring at 500 rpm. The reaction products were centrifuged at 5000 rpm, washed three times with ethanol, and dried at 70 °C for 8 h in a vacuum oven to obtain the SiO_2_ nanoparticles.

After that, the 0.5 g of SiO_2_ nanoparticles and 40 mL of acetonitrile were poured into a 250 mL three-neck flask with a reflux condenser. Then, 0.2 g of MBAA and 0.5 g of AM were dissolved in the above solution, and the mixture was sonicated for 10 min at 100 Hz at ordinary temperature. Then, an 0.1 g of AIBN was poured into the above mixture and sonicated for 5 min at 100 Hz. The mixed solution was reacted at 90 °C for 15 min under the condition of continuous stirring at 500 rpm and then distilled with a reflux ratio of 2 at 115 °C for 90 min. Then, solvents were distilled out, the reaction products were taken out to dissolve in ethyl alcohol, and the mixture was sonicated for 10 min at 100 Hz. Finally, the products were centrifuged at 5000 rpm, washed three times with deionized water, and dried at 50 °C for 12 h in a vacuum oven to obtain the SiO_2_/P(MBAAm-*co*-AM) polymeric nanoparticles. The schematic illustration of the synthetic process of the SiO_2_/P(MBAAm-*co*-AM) polymeric nanoparticles is shown in Figure 1.

### 2.3. Characterization

The morphology of SiO_2_ nanoparticles was determined by a Hitachi S-4500 scanning electron microscope (SEM, Tokyo, Japan). The size, shape, and surface morphologies of SiO_2_/P(MBAAm-*co*-AM) polymeric nanoparticles were characterized by a JEM-200CX transmission electron microscopy (TEM, Japan Electronics Co., Ltd., Tokyo, Japan). The aqueous solutions of SiO_2_/P(MBAAm-*co*-AM) polymeric nanoparticles with a mass concentration of 1500 mg/L were prepared and placed in a thermostat water bath at 60 °C for 24, 48, and 120 h. Then, the particle size distributions of SiO_2_/P(MBAAm-*co*-AM) polymeric nanoparticles were determined by a Mastersizer 2000 laser particle size analyzer (Worcestershire, UK). The apparent viscosities of SiO_2_/P(MBAAm-*co*-AM) polymeric nanoparticle solutions were measured using a LV-DV2T Brookfield viscometer (Middleboro, MA, USA) at different mass concentrations, various temperatures, and different swelling times.

### 2.4. Microtubule Flow Experiment

The solutions of SiO_2_/P(MBAAm-*co*-AM) polymeric nanoparticles with different mass concentrations (200, 500, and 1000 mg/L) were prepared. Quartz microtubules with the inner diameter of 10, 15, and 20 μm were used as the porous media to carry out the flow experiment for measuring the seepage law of SiO_2_/P(MBAAm-*co*-AM) polymeric nanoparticles. The microtubule flow experiment setup is shown in Figure 2. First, the compression release valve of the nitrogen cylinder was opened to fill the pipeline between the compression release valve and the precise decompression valve with the nitrogen. Then, the steering valve between the gas tank and the liquid storage tank was opened, and the precise decompression valve was adjusted to make nitrogen flow from the nitrogen cylinder. The nitrogen entered the liquid storage tank after passing through the compression release valve, the tertiary gas filter, and the gas tank. The solution of SiO_2_/P(MBAAm-*co*-AM) polymeric nanoparticles in the liquid storage tank was smoothly transported to the quartz microtubule under the action of high-pressure nitrogen, and the fluid velocity was measured through the metering tube connected at the end of the quartz microtubule.

The fluid velocity in the metering tube was determined by the displacement method, shown in Figure 3. The metering tube was installed to pass through the photodiode of the photoelectric micro-flowmeter. A bubble was injected into a certain position of the metering tube to form a gas–liquid interface. The photodiode was triggered when the interface was sensed, and the stopwatch automatically started timing. The photodiode was triggered again when the gas–liquid interface reached the preset displacement, and the stopwatch terminated timing. Thus, the actual flow velocity (*v*) in the microtubule was calculated according to the displacement (*S*) and time (*t*) of fluid measured in the experiment. The actual flow velocity was calculated using Equation (1).
(1)$$v=\frac{S}{t}$$

### 2.5. Profile Control Experiment

The sand packs were used as the porous media to carry out the profile control experiment. The quartz sands with different meshes were packed into the stainless steel holders to prepare the sand packs. The sand packs had a cross-sectional area of 5.31 cm^2^ and a length of 1 m. The sands with suitable meshes (between 100 and 160) were packed into the stainless steel holders to obtain the sand packs with the desired permeability. The pressure measuring points were set at the injection end and every 20 cm on the sand pack. Figure 4 shows a schematic diagram of the profile control experiment. Fresh quartz sands were used to prepare two sand packs at the ambient temperature for ensuring the same wettability of them. The porosity of the sand pack was measured using the weight method. The formation brine was first injected into the sand pack by a syringe pump until the pressures at five measuring points remained stable. The initial permeabilities between the measuring points before the profile control were calculated using Darcy’s Law of single water flow according to the measured stable pressures. Then, the solution of SiO_2_/P(MBAAm-*co*-AM) polymeric nanoparticles with a concentration of 1500 mg/L was injected into the sand pack. Finally, after a slug (0.4 PV, pore volume) of SiO_2_/P(MBAAm-*co*-AM) polymeric nanoparticle solution was injected, the followed waterflood was carried out using the formation brine until the pressures at the measuring points stabilized. The final permeabilities between the measuring points after the profile control were obtained by the same calculation method. The data of the pressures during the experiments were recorded. All the sand pack experiments were carried out at 60 °C at an injection rate of 0.5 mL/min.

### 2.6. Microscopic Flooding Experiment

An etched glass micromodel was used as the porous media to conduct the microscopic flooding experiment. According to the real pore structure of the natural core from Daqing Oil Field, a pore network was etched on a glass plate by the photochemical method. The etched glass plate was sintered with another smooth glass plate to fabricate the etched glass micromodel. Small holes were drilled at the two opposite corners of the smooth glass plate to simulate the injection and production wells of the reservoir. The size of the etched glass micromodel was 6.5 cm × 6.5 cm, in which the pore area was 4.0 cm × 4.0 cm with the inner diameters of the pores ranging from 10 to 100 μm. The etched glass micromodel used in this study had three parallel fractures, as shown in Figure 5a. The micromodel was properly cleaned using deionized water, dried at 150 °C for 24 h, and weighed. The dried micromodel was vacuumed, saturated with the formation brine, and weighed again. The porosity of the micromodel was obtained using the weight method. The crude oil was injected into the micromodel to establish the initial oil saturation. The micromodel saturated with crude oil was aged for 24 h to achieve the desired wettability at 60 °C. After that, the synthetic formation water was injected into the micromodel. A slug (0.4 PV) of SiO_2_/P(MBAAm-*co*-AM) polymeric nanoparticle solution with a concentration of 1500 mg/L was injected at waterflood residual oil saturation, followed by an extended waterflood. The microscopic flooding experiment was conducted at 60 °C. The injection rate in the microscopic flooding experiment was 8 μL/min. The images of the micromodel during the experiment were constantly captured. Figure 6 shows the schematic diagram of the microscopic flooding experiment. To calculate the oil saturation from images, the original images of the residual oil were sharpened to strengthen the contrast, and then a program developed using MATLAB was used to analyze the sharpened images [30]. The water, oil, and glass phases in the images were distinguished using the gray thresholds given by the program. Finally, the area ratio of oil to pore space was calculated to obtain the oil saturation.

## 3. Results and Discussion

### 3.1. Characterization of SiO_2_/P(MBAAm-co-AM) Polymeric Nanoparticles

The SEM measurement was carried out to characterize the micrographs of SiO_2_ nanoparticles, as shown in Figure 7a. The SiO_2_ nanoparticles were spherical with a diameter of about 200 nm. The morphologies of SiO_2_/P(MBAAm-*co*-AM) polymeric nanoparticles were detected using TEM, as shown in Figure 7b. The SiO_2_/P(MBAAm-*co*-AM) polymeric nanoparticles were still spherical and comprised a shell of polymer and a core of SiO_2_ nanoparticles. The diameters of SiO_2_/P(MBAAm-*co*-AM) polymeric nanoparticles were in the range of 260–300 nm, and the thickness of the polymeric layer ranged from 30 to 50 nm.

Dynamic light scattering (DLS) was employed to measure the particle size distribution curves of SiO_2_ nanoparticles and SiO_2_/P(MBAAm-*co*-AM) polymeric nanoparticles at swelling times of 24, 48, and 120 h using a laser particle size analyzer, as shown in Figure 8a–d. The particle size distribution of SiO_2_ nanoparticles ranged from 100 to 400 nm, and its medium size (*D*_50_) was 200.64 nm. The particle size distributions of SiO_2_/P(MBAAm-*co*-AM) polymeric nanoparticles were in the ranges of 300–1400, 400–2000, and 500–2500 nm at swelling times of 24, 48, and 120 h, respectively. The SiO_2_/P(MBAAm-*co*-AM) polymeric nanoparticles with particle sizes of 630.96, 891.25, and 1258.93 nm at swelling times of 24, 48, and 120 h were the highest proportions of 16.35%, 14.61%, and 12.47%, respectively. Thus, it is clear that the particle sizes changed from the nano to micron-scale due to the swelling caused by the water absorption, and the particle size distributions moved towards larger sizes with increasing swelling time. In addition, the particle size distribution curves at swelling times of 24, 48, and 120 h were analyzed to obtain the medium sizes of SiO_2_/P(MBAAm-*co*-AM) polymeric nanoparticles. It was found that the medium size increased with an increase in the swelling time. As the swelling time increased from 24 to 120 h, the medium sizes increased from 584.45 to 1142.61 nm.

The apparent viscosities of SiO_2_/P(MBAAm-*co*-AM) polymeric nanoparticle solutions versus mass concentration, temperature, and swelling time were determined at a shear rate of 7.34 s−1 using a viscometer, as shown in Figure 9a–c. As shown in Figure 9a, the apparent viscosities of SiO_2_/P(MBAAm-*co*-AM) polymeric nanoparticle solution increased gradually with an increase in the mass concentration. Figure 9b shows that the apparent viscosities of SiO_2_/P(MBAAm-*co*-AM) polymeric nanoparticle solution decreased with an increase in temperature. As shown in Figure 9c, the apparent viscosities of SiO_2_/P(MBAAm-*co*-AM) polymeric nanoparticle solution increased slowly with an increase in the swelling time. After a swelling time greater than 200 h, the apparent viscosities of SiO_2_/P(MBAAm-*co*-AM) polymeric nanoparticle solution remained stable. Compared with the polymer solution, the mass concentration, temperature, and swelling time had little influence on the apparent viscosities of SiO_2_/P(MBAAm-*co*-AM) polymeric nanoparticle solutions, which is beneficial to improving oil recovery in the complex reservoir.

### 3.2. Microtubule Flow Characteristics

The microtubule flow experiments were carried out to measure the flow velocity of SiO_2_/P(MBAAm-*co*-AM) polymeric nanoparticle solution at constant pressure. The values of the flow velocity were determined three times at every pressure, and then the average flow velocities were calculated. The plots of the flow velocity of SiO_2_/P(MBAAm-*co*-AM) polymeric nanoparticle solutions in different microtubules versus the pressure gradient, at mass concentrations of 200, 500, and 1000 mg/L, are shown in Figure 10. As shown in Figure 10a, the flow velocity of SiO_2_/P(MBAAm-*co*-AM) polymeric nanoparticle solutions decreased rapidly with the decreasing inner diameter of the microtubule at the same pressure gradient. On the one hand, the volume force decreased with a decrease in the inner diameter of the microtubule, while the intermolecular force between the solid surface and nanoparticles gradually increased until the intermolecular force exceeded the volume force [31]. On the other hand, the viscous force inside the liquid also increased with the decreasing inner diameter of the microtubule [32]. Therefore, the above two actions increased the flow resistance of the fluid in the microtubule, resulting in a decrease in the flow velocity. In addition, the flow velocity of SiO_2_/P(MBAAm-*co*-AM) polymeric nanoparticle solutions increased with the increasing pressure gradient in the same microtubule. The flow velocity versus the pressure gradient basically followed a linear relationship. However, at a low-pressure gradient, the flow velocity versus the pressure gradient deviated from the linear relationship and showed the characteristic of the nonlinear flow. The smaller the inner diameter of a microtubule is, the stronger the nonlinearity is. This indicates that the fluid flow had a prominent scale effect at a low-pressure gradient. The decrease of the inner diameter of the microtubule enhanced the intermolecular force between the fluid molecules and solid surfaces and fundamentally changed the exchange of mass, momentum, and energy, as well as the flow state of the fluid in the microtubules [33,34]. When the same experiments were carried out using SiO_2_/P(MBAAm-*co*-AM) polymeric nanoparticle solutions with mass concentrations of 500 and 1000 mg/L, it was found that the flow velocity versus the pressure gradient had a similar change law. However, with an increase in the number of SiO_2_/P(MBAAm-*co*-AM) polymeric nanoparticles in the solution, the fluid had a stronger interaction with the inner wall of the microtubule to create a greater flow resistance, resulting in a larger reduction of the flow velocity at the same pressure gradient.

The plots of the flow velocity of SiO_2_/P(MBAAm-*co*-AM) polymeric nanoparticle solutions with different mass concentrations versus the pressure gradient, in different microtubules with the inner diameters of 10, 15, and 20 μm, are shown in Figure 11. The flow velocity of SiO_2_/P(MBAAm-*co*-AM) polymeric nanoparticle solutions decreased gradually with the increase of the mass concentration at the same pressure gradient. As the mass concentration of SiO_2_/P(MBAAm-*co*-AM) polymeric nanoparticles increased, the flow resistance increased gradually due to the increasing force between the solid surface and nanoparticles and the increasing viscous force inside the fluid [35,36]. In addition, it was found that the flow velocity of deionized water was not the highest at the same pressure gradient, but close to that of the SiO_2_/P(MBAAm-*co*-AM) polymeric nanoparticle solution with a mass concentration of 1000 mg/L. This is because a small amount of the surfactant used in the preparation of SiO_2_/P(MBAAm-*co*-AM) polymeric nanoparticles remained on the surfaces of the nanoparticles. The residual surfactant changed the wettability of the microtubule wall and reduced the flow resistance of fluid in the microtubule. Therefore, the flow velocity of SiO_2_/P(MBAAm-*co*-AM) polymeric nanoparticle solution was greater than that of deionized water at mass concentrations less than 1000 mg/L. However, with the increase of mass concentration, the flow velocity of SiO_2_/P(MBAAm-*co*-AM) polymeric nanoparticle solution decreased gradually due to the increasing flow resistance until it was less than that of deionized water. As shown in Figure 11c, the flow velocities of SiO_2_/P(MBAAm-*co*-AM) polymeric nanoparticle solution at mass concentrations of 2000 and 3000 mg/L could not be measured in the microtubule with an inner diameter of 10 μm, indicating that the scale effect had a significant influence on the fluid flow.

### 3.3. Profile Control Characteristics

The SiO_2_/P(MBAAm-*co*-AM) polymeric nanoparticles swelled for 24 and 120 h at 60 °C and then were used to conduct the profile control experiments in the sand packs. The pressures at five measuring points in the sand packs were determined during the experiments, and then the permeabilities between the measuring points were calculated using Darcy’s Law of single water flow. The plots of the pressure and permeability versus the injection time are shown in Figure 12. As shown in Figure 12a, after the SiO_2_/P(MBAAm-*co*-AM) polymeric nanoparticles swelled for 24 h were injected, the pressures (*P*_1_ and *P*_2_) at the first and second measuring points increased slightly. The *P*_1_ and *P*_2_ began to rise faster after commencing the extended waterflood, with the maximum increase greater than 100 kPa. At this time, the pressures (*P*_3_ and *P*_4_) at the third and fourth measuring points rose slowly. After about 500 min, the *P*_3_ and *P*_4_ started to rise sharply. After about 1000 min, the pressure (*P*_5_) at the fifth measuring point began to rise gradually. This indicates that the pressure change was not obvious during the injection of SiO_2_/P(MBAAm-*co*-AM) polymeric nanoparticles due to their small particle sizes. The particle sizes of SiO_2_/P(MBAAm-*co*-AM) polymeric nanoparticles increased gradually with the increase of the injection time during the extended waterflood because they continued to swell in the sand packs. Thus, the water flow channels were blocked, resulting in a rise of the pressures at the first and second measuring points. With the increase of the injection pressure, some SiO_2_/P(MBAAm-*co*-AM) polymeric nanoparticles were pushed behind the third, fourth, and fifth measuring points to form the effective plugging between the measuring points, resulting in the increase of the pressures at the third, fourth, and fifth measuring points, one after another.

As shown in Figure 12b, due to the lack of effective plugging, the permeability (*K*_12_) between the first and second measuring point had only a small decrease in the initial stage depending on the retention of SiO_2_/P(MBAAm-*co*-AM) polymeric nanoparticles. After about 500 min, the swelled SiO_2_/P(MBAAm-*co*-AM) polymeric nanoparticles formed the effective plugging between the first and second measuring point, and the *K*_12_ decreased rapidly. Then, under pressure, the SiO_2_/P(MBAAm-*co*-AM) polymeric nanoparticles successively broke through and migrated into the subsequent measuring points. The effective plugging was formed in the subsequent region of the sand pack, resulting in a significant reduction of the permeabilities (*K*_23_, *K*_34_, *K*_45_, and *K*_5o_) between the second and third measurement points, the third and fourth ones, the fourth and fifth ones, and the fifth one and outlet of the sand pack. Finally, under the continuous impact of the extended waterflood, some SiO_2_/P(MBAAm-*co*-AM) polymeric nanoparticles broke through from the original plugging area to the next one, and even flowed out from the outlet end of the sand pack, resulting in an increase in the permeabilities between the measuring points. The relative change rates of the permeabilities between the measuring point are shown in Table 1. As shown in Table 1, the permeabilities between the measuring point all decreased before and after the profile control in the sand pack. The SiO_2_/P(MBAAm-*co*-AM) polymeric nanoparticles formed strong plugging in the relatively high permeability area, and the relative change rate of the permeability reached greater than 79%. This indicates that if the SiO_2_/P(MBAAm-*co*-AM) polymeric nanoparticles had a slow expansion speed in the reservoir near the well, it was easy for them to enter into the deep formation with the water flow and could effectively block the high permeability area with the gradual expansion.

As shown in Figure 12c, compared with the SiO_2_/P(MBAAm-*co*-AM) polymeric nanoparticles swelled for 24 h, the *P*_1_, and *P*_2_ increased rapidly due to the injection of the SiO_2_/P(MBAAm-*co*-AM) polymeric nanoparticles swelled for 120 h. The increases in the pressures were more than 20 kPa during the injection of the fully swelled SiO_2_/P(MBAAm-*co*-AM) polymeric nanoparticles. After the extended waterflood started, the *P*_1_ and *P*_2_ continued to rise, and the *P*_3_ also increased quickly. After about 300 min, the *P*_4_ and *P*_5_ started to increase. It can be seen that the fully swelled SiO_2_/P(MBAAm-*co*-AM) polymeric nanoparticles form the effective plugging between the first and third measuring points in the initial stage of the injection, resulting in a significant increase in pressure. With the rapid increase of the *P*_1_ and *P*_2_, the SiO_2_/P(MBAAm-*co*-AM) polymeric nanoparticles entered the rear measuring areas at a faster speed and formed the effective plugging in a short time. This indicates that the swelling time has an important influence on the profile control performance of SiO_2_/P(MBAAm-*co*-AM) polymeric nanoparticles. As shown in Figure 12d, for the low permeability areas, the SiO_2_/P(MBAAm-*co*-AM) polymeric nanoparticles could greatly reduce their permeabilities, but they were not completely blocked. For high permeability areas, the SiO_2_/P(MBAAm-*co*-AM) polymeric nanoparticles could also significantly reduce their permeabilities by plugging the high permeability strips and large pores. Moreover, the significant reduction of the *K*_45_ shows that the fully swelled SiO_2_/P(MBAAm-*co*-AM) polymeric nanoparticles could still effectively block the high permeability area after being sheared in the low permeability area. As shown in Table 1, the relative change rate of the highest permeability (*K*_45_) reached greater than 90%. It is derived that the SiO_2_/P(MBAAm-*co*-AM) polymeric nanoparticles had good shear resistance and secondary plugging ability. Therefore, the SiO_2_/P(MBAAm-*co*-AM) polymeric nanoparticles could effectively change the water phase permeability in the deep formation and enhance oil recovery by increasing the sweep efficiency of water.

### 3.4. Microscopic Flooding Characteristics

The micromodel with three fractures was applied to study the profile controlling effect and the ability of enhanced oil recovery of the SiO_2_/P(MBAAm-*co*-AM) polymeric nanoparticle solution, as shown in the photographs of Figure 13. The flow direction was labeled with arrows in the photographs. Figure 13a shows the initial oil saturation contained in the micromodel. Figure 13b shows the oil distribution at the end of the waterflood. It was seen that oil only in the fractures and some large channels was swept by water. There was a lot of residual oil in the un-swept region of the micromodel. Figure 13c shows the oil distribution after the injection of the SiO_2_/P(MBAAm-*co*-AM) polymeric nanoparticles. The SiO_2_/P(MBAAm-*co*-AM) polymeric nanoparticles selectively flowed first into the fractures and large channels, then switched into the small channels with increasing flow resistance in the fractures and large channels. The plugging of the fractures and large channels forced the SiO_2_/P(MBAAm-*co*-AM) polymeric nanoparticles to enter a small portion of the un-swept small channels. Therefore, it is easy to determine that a swept strip was formed in the upper right corner of the micromodel due to the plugging of the SiO_2_/P(MBAAm-*co*-AM) polymeric nanoparticles for the fractures and macro-pores. Figure 13d shows the oil distribution at the end of an extended waterflood. The water was diverted to a large number of the un-swept small channels due to the plugging of the fractures and large channels during the extended waterflood. The SiO_2_/P(MBAAm-*co*-AM) polymeric nanoparticles could plug dynamically and alternatively the pore throats of different sizes to regulate the flow between the small and large channels. Eventually, the extended waterflood could recover the abundant residual oil in the un-swept small channels.

To quantitatively analyze the residual oil and enhanced oil recovery (EOR), the micromodel was divided into three sections, namely the main channel area, transition region, and boundary zone, to calculate the residual oil saturation and EOR, respectively, as shown in Table 2. The distribution proportions of the residual oil in the boundary zone, transition region, and main channel area at the end of the waterflood show that the water preferentially went through the dominant channel to form the connection between the inlet and outlet during the waterflood. When the SiO_2_/P(MBAAm-*co*-AM) polymeric nanoparticles were injected, they preferentially entered the fractures in the main channel area, and the flow resistance increased. The subsequent fluid was diverted to the transition area and boundary zone, and the swept volume was improved so that the EORs in the boundary zone and transition region were greater than that in the main channel area. The high flow resistance in the main channels reduced the mobility of water and further improved the swept volume of water during the extended waterflood. Finally, more residual oil was driven from the micromodel, and the EOR could reach 20.71% by the SiO_2_/P(MBAAm-*co*-AM) polymeric nanoparticles in the whole micromodel.

In addition, the flow characteristics of SiO_2_/P(MBAAm-*co*-AM) polymeric nanoparticles in the micromodel are shown in Figure 14. As shown in Figure 14a, in the fracture, a large number of SiO_2_/P(MBAAm-*co*-AM) polymeric nanoparticles gathered together to form a dense network structure that brought about the reticular retention so that the flow resistance of the fluid was improved, and then the subsequent fluid was forced to enter the nearby channels. As shown in Figure 14b, some SiO_2_/P(MBAAm-*co*-AM) polymeric nanoparticles gathered together in the form of a loose network and flow slowly in the large channel. When these SiO_2_/P(MBAAm-*co*-AM) polymeric nanoparticles reached the throat, the loose network structure was destroyed, and the dispersed SiO_2_/P(MBAAm-*co*-AM) polymeric nanoparticles slowly passed through the throat to enter the next channel. As shown in Figure 14c, the SiO_2_/P(MBAAm-*co*-AM) polymeric nanoparticles were elastic and passed through the small channel by deformation under a certain pressure. However, when the pressure was not enough to overcome the resistance of the small channel, the SiO_2_/P(MBAAm-*co*-AM) polymeric nanoparticles stayed in and plugged the channel. Figure 14d–i shows the flow states of SiO_2_/P(MBAAm-*co*-AM) polymeric nanoparticles in three positions at different times. It is seen that the SiO_2_/P(MBAAm-*co*-AM) polymeric nanoparticles could flow in a loose network structure in the channels and recover this structure in the next channels after dispersedly passing through the throat. This indicates that the SiO_2_/P(MBAAm-*co*-AM) polymeric nanoparticles could enter the deep part of the reservoir to realize the step-by-step profile control, which reduced the mobility of the subsequent fluid, changed its flow direction, and enhanced oil recovery by improving the swept volume.

## 4. Conclusions

In this work, we used quartz microtubules, sand packs, and etched glass micromodels as the porous media to investigate the seepage law and migration characteristics of a SiO_2_/P(MBAAm-*co*-AM) polymeric nanoparticle solution. We expect that this work will be conducive to better understanding of the flow characteristics of SiO_2_/P(MBAAm-*co*-AM) polymeric nanoparticles in porous media. The major conclusions are drawn as follows:(1)The flow of SiO_2_/P(MBAAm-*co*-AM) polymeric nanoparticle solutions show obvious nonlinear characteristic at a low-pressure gradient in a single channel. The smaller the channel size is and the larger the mass concentration is, the stronger the nonlinearity of the fluid flow is. The fluid flow has a prominent scale effect at a low-pressure gradient, and there should be an optimal injection mass concentration of the SiO_2_/P(MBAAm-*co*-AM) polymeric nanoparticles for the channels with different sizes.(2)After entering the reservoir, SiO_2_/P(MBAAm-*co*-AM) polymeric nanoparticles block the main flow channels, reduce the permeability of the water phase, and form effective plugging at different permeability areas. SiO_2_/P(MBAAm-*co*-AM) polymeric nanoparticles first accumulate and block the large channel, and then reaccumulate in the next large channel after breaking through the pore throat to fulfill the step-by-step profile control. The swelling time has a large influence on the profile control of the SiO_2_/P(MBAAm-*co*-AM) polymeric nanoparticles. When the swelling time increases from 24 to 120 h, the maximum change rate of the permeability induced by the profile control increases from 79.35% to 93.00%.(3)The SiO_2_/P(MBAAm-*co*-AM) polymeric nanoparticles gather together with a dense network structure in the fracture to form the retention, flow slowly with the form of a loose network in the large channel, and either pass through the small channel by elastic deformation or are directly detained in the small channel. Therefore, the SiO_2_/P(MBAAm-*co*-AM) polymeric nanoparticles can block dynamically and alternatively the pore throats of different sizes to regulate the fluid flow between the small and large channels. The SiO_2_/P(MBAAm-*co*-AM) polymeric nanoparticles can significantly give rise to the fluid diversion, effectively improve the sweep efficiency, and recover more residual oil in the un-swept small channels. SiO_2_/P(MBAAm-*co*-AM) polymeric nanoparticles can achieve an enhanced oil recovery of 20.71%.

## Figures and Tables

**Figure 1 polymers-14-01803-f001:**
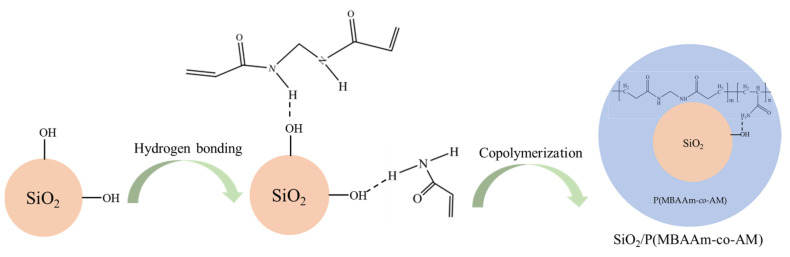
Schematic illustration of the synthetic process of the SiO_2_/P(MBAAm-*co*-AM) polymeric nanoparticles.

**Figure 2 polymers-14-01803-f002:**
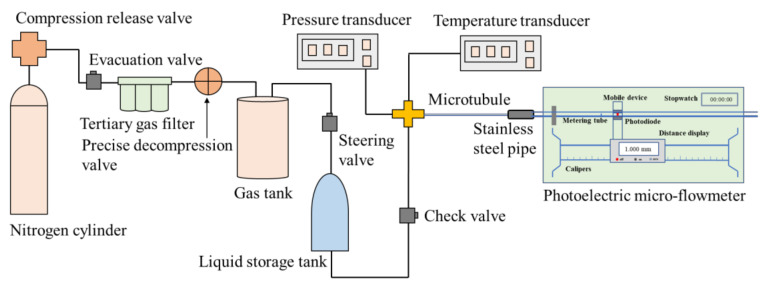
Schematic diagram of the microtubule flow experiment. The experimental equipment comprised a nitrogen cylinder to provide driving gas, a tertiary gas filter to filtrate impurities in gas, a gas tank to store the filtered gas, a liquid storage tank to store the aqueous solution of SiO_2_/P(MBAAm-*co*-AM) polymeric nanoparticles, a pressure transducer to determine the driving pressure, a temperature transducer to determine the ambient temperature, and a photoelectric micro-flowmeter to measure the fluid velocity in the microtubule.

**Figure 3 polymers-14-01803-f003:**
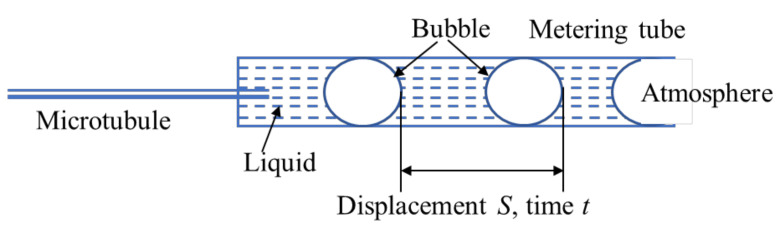
Schematic diagram of flow measurement by displacement method in the metering tube.

**Figure 4 polymers-14-01803-f004:**
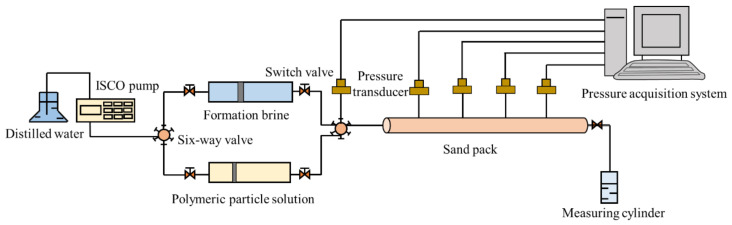
Schematic diagram of the profile control experiment. The experiment set-up consisted of a sand pack, a syringe pump to inject fluid at a constant flow rate, a measuring cylinder to collect the residual liquid flowing from the outlet of the sand pack, and five pressure transducers that were connected to a computer for continuous recording of the pressures at different positions of the sand pack.

**Figure 5 polymers-14-01803-f005:**
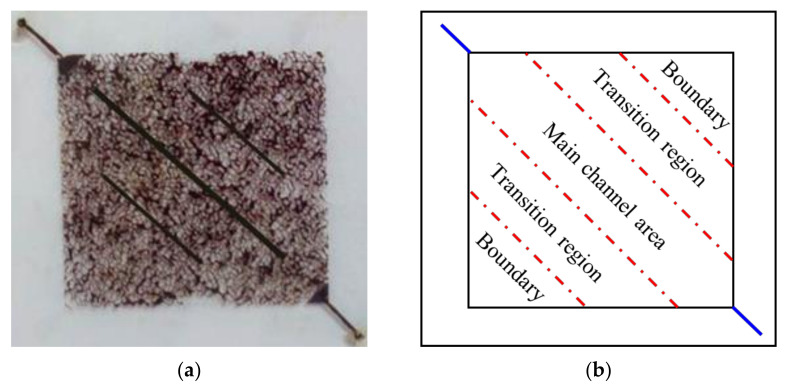
Photograph (**a**) of the etched glass micromodel saturated with crude oil, and diagrammatic sketch (**b**) of its regional division.

**Figure 6 polymers-14-01803-f006:**
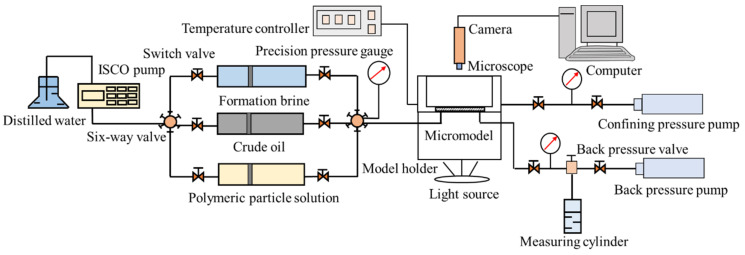
Schematic diagram of the microscopic flooding experiment. The experimental setup included the flooding and image capturing systems. The flooding system comprised a syringe pump; three accumulators to store the formation brine, crude oil, and polymeric particles solution; a stainless steel model holder that was a hollow cylinder with glass windows at the bottom and top; a micromodel that was clamped horizontally by the thick glass cover inside the model holder; a temperature controller to regulate the temperature inside the model holder; a confining pressure pump to load the pressure around the micromodel through injecting water into the model holder; a backpressure valve to simulate the actual formation pressure; and a back pressure pump to load the reverse pressure for the backpressure valve. The image capturing system consisted of a light source to penetrate vertically through the micromodel, a camera, a microscope, and a computer.

**Figure 7 polymers-14-01803-f007:**
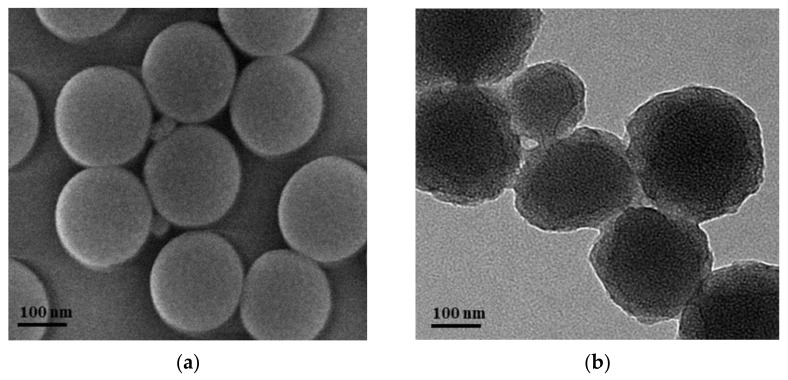
SEM image of SiO_2_ nanoparticles (**a**), and TEM image of SiO_2_/P(MBAAm-*co*-AM) polymeric nanoparticles (**b**).

**Figure 8 polymers-14-01803-f008:**
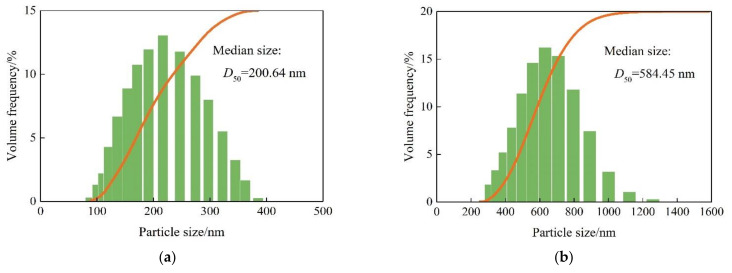
Particle size distribution curves of SiO_2_ nanoparticles (**a**) and SiO_2_/P(MBAAm-*co*-AM) polymeric nanoparticles swelled for 24 (**b**), 48 (**c**), and 120 h (**d**) at a temperature of 60 °C and a mass concentration of 1500 mg/L.

**Figure 9 polymers-14-01803-f009:**
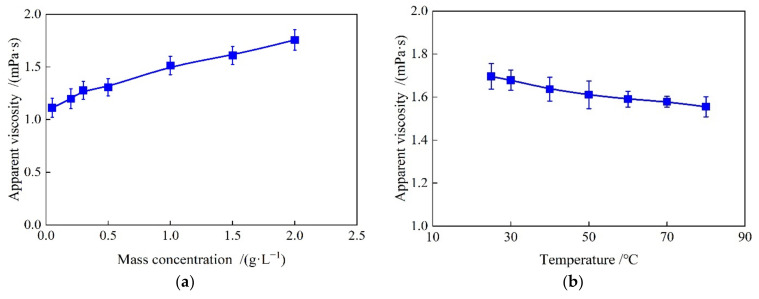
Apparent viscosity of SiO_2_/P(MBAAm-*co*-AM) polymeric nanoparticle solutions versus mass concentration (**a**) at a temperature of 60 °C and a swelling time of 120 h, versus temperature (**b**) at a mass concentration of 1500 mg/L and a swelling time of 120 h, and versus swelling time (**c**) at a temperature of 60 °C and a mass concentration of 1500 mg/L.

**Figure 10 polymers-14-01803-f010:**
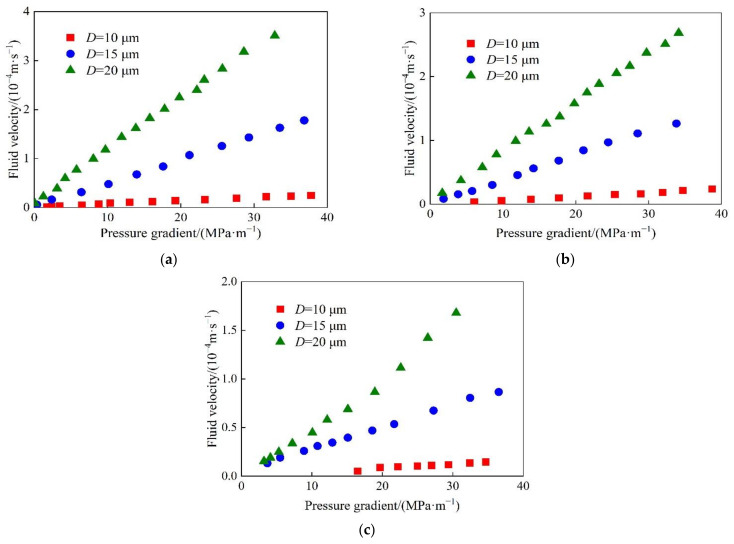
Flow velocity of SiO_2_/P(MBAAm-*co*-AM) polymeric nanoparticle solutions in different microtubules versus the pressure gradient at mass concentrations of 200 (**a**), 500 (**b**), and 1000 (**c**) mg/L.

**Figure 11 polymers-14-01803-f011:**
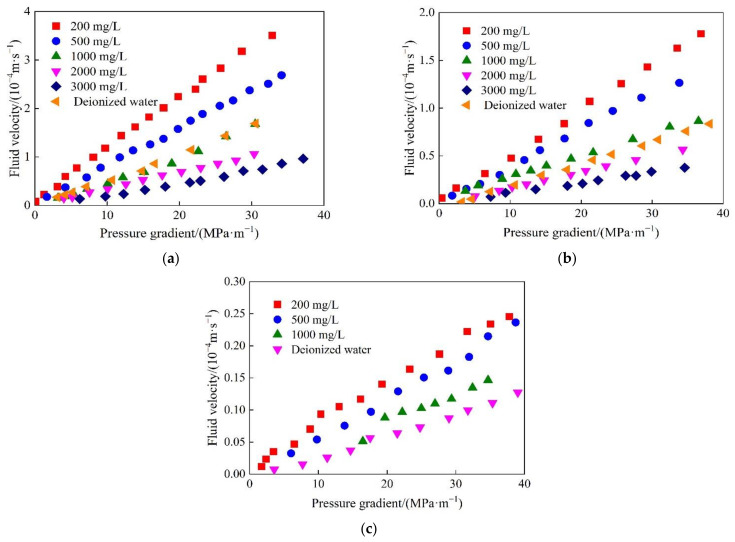
Flow velocity of SiO_2_/P(MBAAm-*co*-AM) polymeric nanoparticle solutions at different mass concentrations versus the pressure gradient in the microtubules with the inner diameters of 20 (**a**), 15 (**b**), and 10 (**c**) μm.

**Figure 12 polymers-14-01803-f012:**
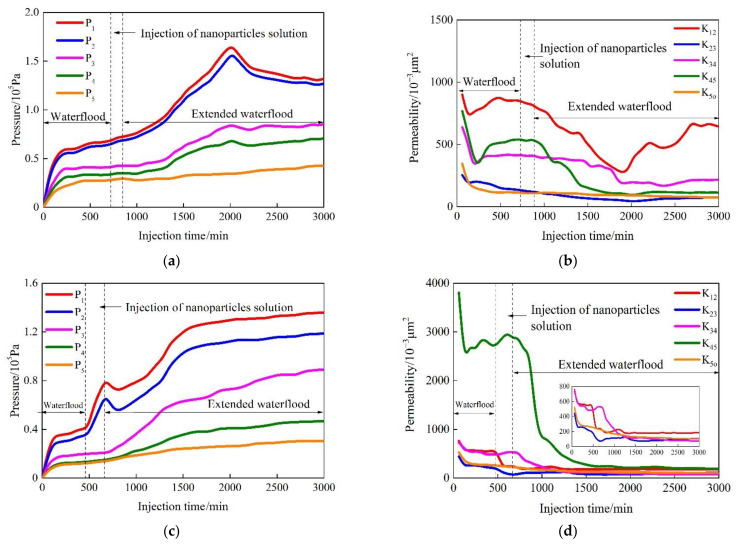
Variation curves of the pressure and permeability with the injection time in different locations of the sand packs before and after the injections of SiO_2_/P(MBAAm-*co*-AM) polymeric nanoparticles swelled for 24 h (**a**,**b**) and swelled for 120 h (**c**,**d**).

**Figure 13 polymers-14-01803-f013:**
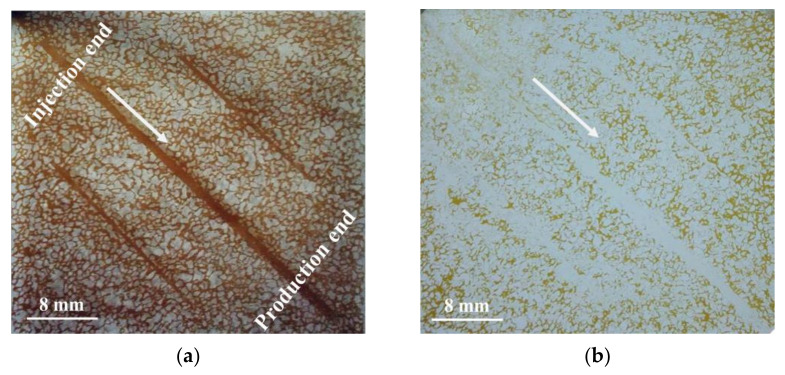
Distribution of the crude oil at different stages of the microscopic flooding experiment: (**a**) initial oil saturation; (**b**) residual oil after waterflood; (**c**) residual oil after the injection of SiO_2_/P(MBAAm-*co*-AM) polymeric nanoparticles; and (**d**) residual oil after extended waterflood.

**Figure 14 polymers-14-01803-f014:**
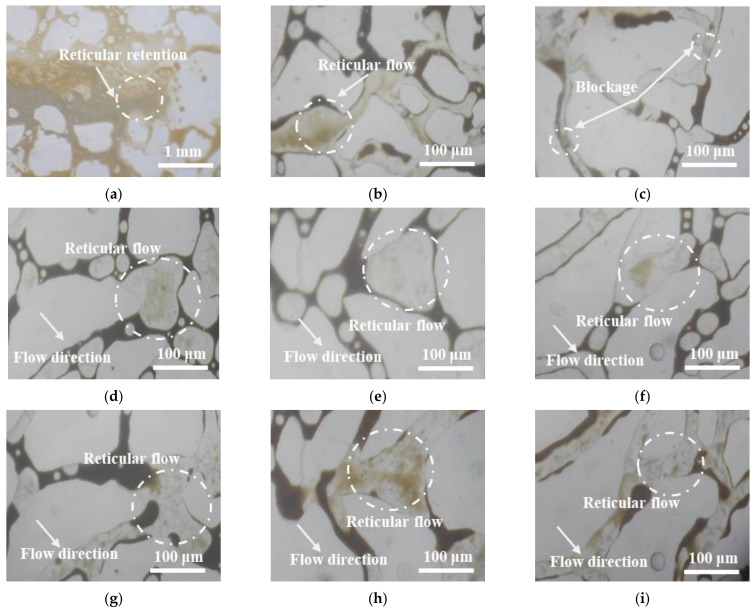
Flow characteristics of SiO_2_/P(MBAAm-*co*-AM) polymeric nanoparticles at different pore channels in the micromodel: (**a**) fracture; (**b**) large channel; (**c**) small channel; (**d**,**g**), (**e**,**h**), and (**f**,**i**) are photos of three positions at different times.

**Table 1 polymers-14-01803-t001:** Relative change rates of the permeabilities between the measuring points in the sand packs before and after the profile control.

No.	Swelled for 24 h			Swelled for 120 h		
	Initial Permeability (10^−3^ μm^2^)	Final Permeability (10^−3^ μm^2^)	Relative Change Rate (%)	Initial Permeability (10^−3^ μm^2^)	Final Permeability (10^−3^ μm^2^)	Relative Change Rate (%)
L12	847.85	644.44	23.99	572.99	438.31	23.51
L23	132.75	75.29	43.28	193.92	106.78	44.94
L34	416.72	217.41	47.83	476.65	73.89	84.50
L45	540.88	111.68	79.35	2732.27	191.29	93.00
L5o	112.01	73.51	34.37	257.37	103.51	59.78

**Table 2 polymers-14-01803-t002:** Enhanced oil recovery at different sections of the micromodel.

Model Area	Residual Oil Saturation (%)			Enhanced Oil Recovery (%)
	Waterflood	Polymeric Nanoparticle Flooding	Extended Waterflood	Polymeric Nanoparticles Flooding and Extended Waterflood
Main channel	40.32	35.43	21.87	18.45
Transition	49.83	38.51	28.97	20.86
Boundary	59.24	49.64	36.44	22.80
Whole model	49.80	41.19	29.09	20.71

## Data Availability

The data that support the findings of this study are available from the corresponding authors upon reasonable request.

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
