# Peer review of "Experimental Research on Seepage Law and Migration Characteristics of Core-Shell Polymeric Nanoparticles Dispersion System in Porous Media"

_polymers, 2022, doi:10.3390/polym14091803_

Round 1
Reviewer 1 Report
- The abstract of the manuscript lacks the results of morphology and particle size distribution.
- The introduction is very long and exhaustive; it can be resumed.
- Introduction part lacks the uses and benefits of the present study. No any novelty statement is there in the last paragraph of the introduction.
- Line 134: The value of total dissolved solids (TDS) should be mentioned in parts per million (ppm).
- Correct the symbol of degree centigrade throughout the manuscript.
- Line 140: Replace the word combine with any other suitable word like mixed or something else.
- Section 2.2, What was the conditions of continuous magnetic stirring? i.e. at what rpm? Similarly, what was the conditions/parameter for sonication and centrifugation?
- What could be the possible reason, that to obtain the SiO2 nanoparticles, it was vacuum dried at 70 0C while it was done at 50 0C to get the SiO2/P(MBAAm-co-AM) polymeric nanoparticles.
- At line 190: Mention it as “The actual flow velocity was calculated by following the Equation (1).
- Line 278: How the thicknesses of the polymeric layer were determined and found in the range of 30 to 50 nm?
- The size distribution curve of SiO2 nanoparticles should also be reported.
- How many time the apparent viscosities were determined for one sample? The viscosities should be determined multiple times and the curve represented in Figure 8, must have standard error bar at each point.
- As the theme of the present manuscript is the development of somewhat new materials, therefore, the synthesized SiO2 nanoparticles and the SiO2/P(MBAAm-co-AM) polymeric nanoparticles must be characterized by means of DSC, FTIR and XRD also.
- The applications/uses or benefits of the present investigation must be explained either in the results and discussion section or in the conclusion section.
Reviewer 2 Report
In this manuscript, the authors investigated the seepage law, migration characteristics and EOR ability of the SiO2/P(MBAAm-co-AM) polymeric nanoparticle solutions. The manuscript is well written, I have some minor suggestions.
- I suggest the authors to add a scheme showing the chemical structures of SiO2/P(MBAAm-co-AM) to help the readers understand the chemistry.
- In the introduction section, I suggest the authors to add one or two sentences to introduce the general use of polymeric nanoparticles for different applications and the different kinds of nanoparticles that have been used for EOR. Some papers can be cited: polymeric nanoparticles for energy applications: "Hybrid conjugated polymer/magnetic nanoparticle composite nanofibers through cooperative non-covalent interactions." Nanoscale Advances 2.6 (2020): 2462-2470.; polymeric nanoparticles for medical applications: "Polymeric nanoparticles: the future of nanomedicine." Wiley Interdisciplinary Reviews: Nanomedicine and Nanobiotechnology 8.2 (2016): 271-299.; Other inorganic nanoparticles used for EOR: "Enhanced heavy oil recovery in sandstone cores using TiO2 nanofluids." Energy & Fuels 28.1 (2014): 423-430.
- I suggest the authors to conduct more characterizations to confirm the success synthesis of SiO2/P(MBAAm-co-AM), for examples IR and NMR.
- Figure 10 (c), the x axis should be pressure gradient instead of fluid velocity.
- Table 1, the first column, what does Nr. stand for?
- I suggest the authors to add scale bars to figures 11 and 12.
Round 2
Reviewer 1 Report
Authors have revised the manuscript appropriately.